# More meteorological events that drive compound coastal flooding are projected under climate change

Emanuele Bevacqua [1,2✉], Michalis I. Vousdoukas[3], Giuseppe Zappa[1,4], Kevin Hodges[1], Theodore G. Shepherd[1], Douglas Maraun [2], Lorenzo Mentaschi [3] & Luc Feyen [3]

Compound flooding arises from storms causing concurrent extreme meteorological tides (that is the superposition of storm surge and waves) and precipitation. This flooding can severely affect densely populated low-lying coastal areas. Here, combining output from climate and ocean models, we analyse the concurrence probability of the meteorological conditions driving compound flooding. We show that, under a high emissions scenario, the concurrence probability would increase globally by more than 25% by 2100 compared to present. In latitudes above 40° north, compound flooding could become more than 2.5 times as frequent, in contrast to parts of the subtropics where it would weaken. Changes in extreme precipitation and meteorological tides account for most (77% and 20%, respectively) of the projected change in concurrence probability. The evolution of the dependence between precipitation and meteorological tide dominates the uncertainty in the projections. Our results indicate that not accounting for these effects in adaptation planning could leave coastal communities insufficiently protected against flooding.

[1] Department of Meteorology, University of Reading, Meteorology Building, Whiteknights Road, Earley Gate, Reading RG6 6ET, UK. [2] Wegener Center for Climate and Global Change, University of Graz, Brandhofgasse 5, 8010 Graz, Austria. [3] European Commission, Joint European Research Centre (JRC), Via Enrico Fermi 2749, I-21027 Ispra, Italy. [4] Istituto di Scienze dell'Atmosfera e del Clima, Consiglio Nazionale delle Ricerche (ISAC-CNR), Via Piero Gobetti 101, 40129 Bologna, Italy. ✉email: e.bevacqua@reading.ac.uk

A considerable portion of the global population lives in low-lying coastal areas that are at risk of flooding from sea-level[1–3] and precipitation/river discharge extremes[4–7]. The concurrence or short succession of these two hazards can cause compound flooding that can result in larger impacts than would be caused by the individual hazards acting alone[8–11], as exemplified by recent events in the Shoalhaven estuary (Australia, 2016)[12], Ravenna (Italy, 2015)[13], Cork (Ireland, 2009), and Lymington (United Kingdom, 1999)[14]. Compound flooding occurs when flooding from inland rainfall is enhanced by high meteorological tides that obstruct the gravity-driven drainage of excess fluvial and/or pluvial water into the sea[13,15,16], or when flooding from meteorological tides is amplified by precipitation[8].

Compound flooding has been studied at local to global scale for present climate. Studies of recent mid-latitude events have increased the understanding of the mechanisms driving compound flooding[12,13,17–20]. At a larger scale, present-day compound flood hazard has been assessed for the United States[8], Australia[21,22], and Europe[9,10,14,15,23] by considering co-occurring sea-level and either precipitation or river discharge extremes. Most of these studies used field observations of sea level, which do not cover the entire global coastline[10,24]. Recent advances in ocean modelling have resulted in the generation of sub-daily continuous time-series of sea level with global coverage[1,25], which together with estimates of either precipitation or river discharge enable comprehensive global assessments of present-day compound flood hazard[26,27]. For example, Eilander et al.[28], using hydrodynamical modelling, focused on present-day compound flooding in river deltas highlighting that storm surge exacerbates 1-in-10 year flood levels in 64.0% of the analysed deltas worldwide.

In the future, sea level rise (SLR) resulting, e.g., from thermal expansion and melting of continental glaciers and polar ice sheets, will push mean and extreme sea levels upward[1] and will thereby increase the future compound flood hazard[10,29]. However, meteorological drivers of compound flooding such as extreme precipitation, meteorological tide, and their interplay will also be affected by climate change[1,5,30]. For example, a warmer atmosphere will favour an increase in the atmospheric moisture content, resulting in more intense precipitation extremes in most coastal areas worldwide[5,30]. Changes in storm frequency and intensity will affect meteorological tides, and are expected to result in associated changes in extreme sea level[1]. Therefore, it is likely that the potential for compound flooding will change along with the changes in these driving meteorological processes, beyond the effects driven by mean SLR. This has been shown for Europe's coasts[10], but such information is currently missing for most low-lying coastal areas around the world. The above, in combination with the expected future increase in coastal population, highlights the need for a comprehensive assessment of the meteorological drivers of compound flooding and their response to climate change.

Here, we develop such an assessment. Following a methodology established in previous studies[8,10,24], we analyse the probability of concurring meteorological tide and precipitation extremes near the coast. Although our estimates should not be interpreted as an actual calculation of the flooding[10,24,26,31], Bevacqua et al.[27] have shown that precipitation can provide a reliable estimate of compound flood potential from pluvial effects and in short-sized and medium-sized rivers, i.e. catchment size up to $5–10 \times 10^3$ km², which is where the compound flood risk is the highest. We first assess the present-day (1980–2014) probabilities of concurring meteorological extremes, including an analysis of their seasonality and physical drivers through focusing on storm tracks. Second, we analyse the changes in the compound extremes by the end of the century (2070–2099) compared to the recent past (1970–2004), and highlight areas with the largest trends. Third, we disentangle, quantify, and interpret the contribution of

the meteorological drivers of compound flooding as well as the dependence between them[8,10] to the overall change. Finally, we investigate the uncertainties in the changes and how these are related to those of the meteorological drivers.

## Results

We combined outputs from global climate models (GCMs) and ocean models in order to assess the spatio-temporal dynamics of the meteorological drivers of compound flooding along the global coastline. Daily time series of meteorological tides were obtained as the superposition of storm surges[1] and waves[1,32,33], which were available from ocean model simulations forced with reanalysis data for the observed past[34], and with GCM climate projections of the Coupled Model Intercomparison Project Phase 5 (CMIP5) for estimates of climate change under a high anthropogenic greenhouse gas emissions scenario (RCP8.5)[35]. Given that higher-resolution input data could improve the representation of extreme events[1,25,36], especially for tropical cyclones (TCs)[25,37–39], we improved the representation of TC-driven meteorological tides in the reanalysis-based dataset. Storm surges caused by TCs were forced by dynamically downscaled atmospheric conditions and waves were corrected for TC effects based on satellite altimetry data (see Vousdoukas et al.[1] for more details). However, this procedure was not feasible for CMIP5-based simulations, therefore, despite the overall satisfactory representation of the compound hazard based on CMIP5 models in the present climate (Supplementary Fig. S1), the projected changes in regions subject to high TC activity should be interpreted with caution.

Daily precipitation time series from the reanalysis data and GCMs in the neighbouring coastal zone were aggregated over 3-day windows[8,10,27,40]. Note that at high latitudes, where precipitation often occurs as snow and therefore cannot directly contribute to flooding, the findings should be interpreted with caution. As discussed earlier, by using aggregated precipitation we do not aim at representing the compound flood potential in estuaries of long rivers (catchment $\gtrsim 5–10 \times 10^3$ km²)[27], for which high discharges close to the coast are influenced by several processes over the catchment inland[24,27,41]. However, employing aggregated precipitation allows for considering local-rainfall-driven compound flood and, with some regional exceptions that will be discussed later, compound flood in small-size and medium-size rivers, including small rivers not resolved by large-scale datasets[27].

We define extremes of meteorological tide and aggregated precipitation as events that occur on average once a year in the present climate of the individual models. We assessed the joint return period (or inverse probability) of concurring extremes in present and future climate[42] from the bivariate distribution of meteorological tides and precipitation based on parametric copulas that model the pairs of high values only[8,10].

**Present-day concurrence of extreme precipitation and meteorological tide**. Present-day concurrence probabilities of ocean and inland meteorological extremes, expressed as joint return period, are mapped in Fig. 1 and summarised for the regions considered by the Intergovernmental Panel on Climate Change (IPCC) in Supplementary Table 1. Low joint return periods indicate a higher dependency between extremes in precipitation and meteorological tide, with a return period of 365 years expected under complete independence. The global median return period of 17 years shows that in the majority of coastlines around the world, ocean and inland meteorological extremes are strongly correlated and roughly 20 times more likely to co-occur compared to if they were independent. There is, however, strong spatial variability in

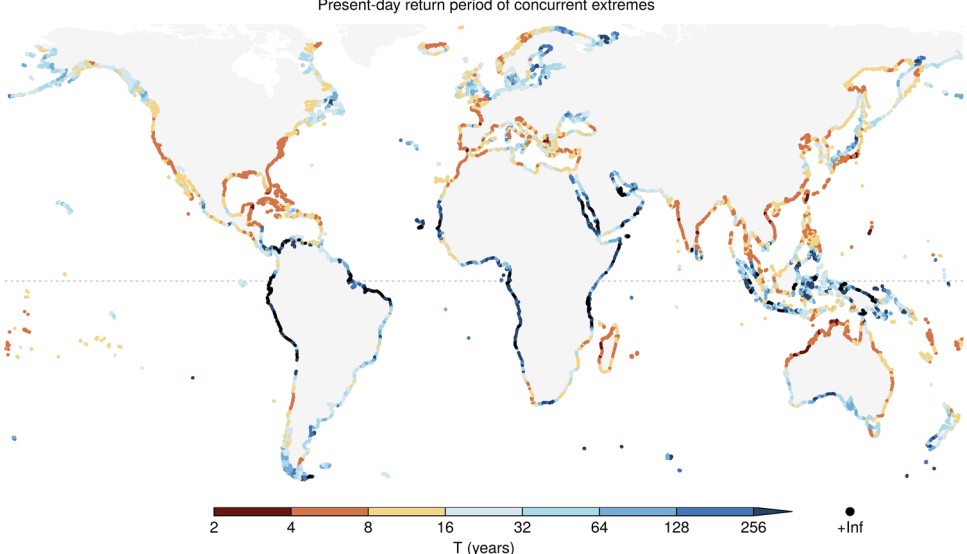

Present-day return period of concurrent extremes

T (years)

**Fig. 1 Present-day return period of concurrent extremes in precipitation and meteorological tide.** Return period (or inverse probability) of co-occurring extremes based on the ERA-Interim data (1980–2014).

concurrence probability, with joint return periods that can vary by several orders of magnitude.

Cyclonic atmospheric flows cause strong surface winds and low atmospheric pressure at the sea surface with consequent high meteorological tides[10,18], but also can lead to intense precipitation. As a result, concurrent extreme precipitation and high meteorological tides are primarily driven by such low-pressure systems (Supplementary Fig. S2)[8,10]. The highest concurrence probability (lowest joint return period) is therefore observed in regions with high tropical (TC) or extratropical (ETC) cyclone activity, such as the United States, eastern Central America, Madagascar, Europe, northern Africa, northern and eastern Australia, India, northern Southeast Asia, China, and Japan (Figs. 1 and 2a, b). In these regions, extreme precipitation and meteorological tides presently coincide every 4–8 years. Off the west coast of Central America and Mexico, TCs are also frequent but they usually travel away from the coast near Central America (Fig. 2b), which results in somewhat higher joint return periods (8–16 years). Overall, concurrence probabilities in the Northern Hemisphere (median return period of 15 years) tend to be higher than in the Southern Hemisphere (23 years), which is consistent with the land distribution relative to the storm tracks (Fig. 2a, b) (e.g., 46% of the coast experiences more than four cyclones per month per 5° spherical cap in the Northern Hemisphere compared to 18% in the Southern Hemisphere). Furthermore, extensive tropical regions with low cyclonic activity exhibit low concurrence probabilities (Figs. 1 and 2a, b).

While cyclones cause high sea levels, high cyclonic activity does not always imply a high probability of concurrent intense precipitation. For example, although cyclones are more frequent over northern Europe than the Mediterranean Sea (Fig. 2a), the two regions show comparable joint return periods (Fig. 1) because the fraction of precipitation extremes caused by cyclones is similar among the two regions[43]. In addition, even with high cyclonic activity and the relevance of cyclones for precipitation extremes, the regional concurrence probability can be low. This is particularly the case when high meteorological tides and precipitation extremes are caused by different cyclone types (e.g. cyclones tracking on different pathways)[14], e.g. in the northern Baltic Sea, western Japan, and western Sardinia (Italy) (Supplementary Fig. S3). We find in general that in areas with a low concurrence probability, coastal and inland meteorological extremes tend to happen in different seasons (see similar spatial distribution of blue areas in Fig. 1 and Supplementary Fig. S4a). This occurs, for example, in large areas of the tropics, indicating that the potential for flooding from concurrent meteorological extremes is limited. It may nevertheless be the case that there is a high risk of flooding from either hazard acting alone; our analysis here is of the compound flood hazard.

The highest number of concurring meteorological tide and precipitation extremes in the tropics tend to occur in the TC season (Fig. 2c). This is during May–November in central America, October–May around Madagascar and northern Australia, April–December around India, and April–January in the Typhoon region. Similarly, at midlatitudes the concurrence frequency peaks around autumn–winter, when the ETC activity is highest[43]. Figure 2d shows the length of the compound season, i.e. the season within which 90% of coincident extremes occurs. For example, in Portugal concurrent extremes tend to occur mostly in December (Fig. 2c) and the season is about 3 months long (Fig. 2d), indicating that most of the concurrent extremes are observed around November–January. The longest season with concurrent extreme events is found along the eastern United States coast (Fig. 2d), where they are caused by both TCs and ETCs[8] which hit the coast in different seasons.

Our findings on the present-day dependence between meteorological tides and precipitation agree in general with those in previous studies[8,10,21,22,24,26,27,40] (Supplementary Discussion). Recently, Eilander et al. (2020)[28] quantified the influence of storm surge on extreme water levels in estuaries for present climate. They did not account for the local pluvial effects considered in the present study, which could further influence the water levels[28]. A direct comparison with the results of Eilander et al.[28] is not straightforward because of different criteria employed for the selection of compound events[28], different thresholds used to define the extremes, and different datasets employed. Overall, the large-scale spatial pattern of compound flooding that we find is similar to that identified by Eilander et al.[28] and previous analyses of the compound flood potential based on river discharge data[24,26,27]. In southern Europe and western South Africa, the compound flood hazard in estuaries is relatively low[28], while we detect a relatively high potential for flooding from concurring precipitation and storm surge extremes (Fig. 1). These deviations may be due to differences in the employed data set rather than to

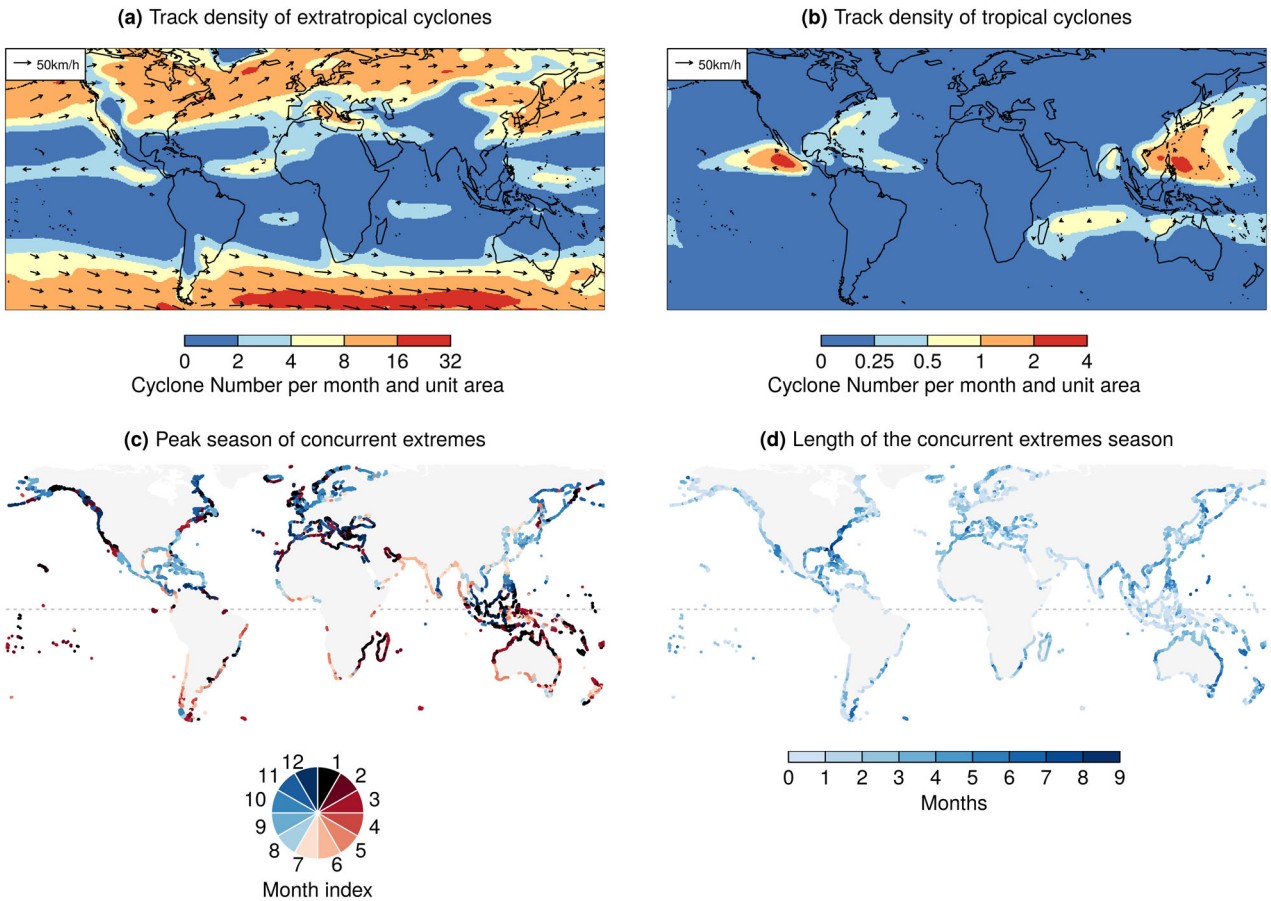

**Fig. 2 Drivers and seasonality of present-day concurrent extremes in precipitation and meteorological tide.** Track density of **a** extratropical cyclones from ERA-Interim and **b** tropical cyclones from IBTrACS (see "Methods" section). Values express the number of cyclones per month per unit area (equivalent to a 5° spherical cap). Arrows show the mean translation speed of cyclones. **c** The month with the highest concurrence of extremes and **d** length of the concurrence season, based on ERA-Interim. The length of the concurrence season was defined as the shortest possible period within which 90% (range defined by the 5–95th percentiles) of concurrent extremes were observed. In **c-d**, extremes are defined based on the 99.5th percentiles.

topographic and hydrological effects[27]. In the next section, we present projections of future compound flooding potential from pluvial and fluvial processes; results about fluvial compound flooding should be considered with caution in regions where we detect rare concurring meteorological extremes, but Eilander et al.[28] identify relatively high fluvial compound flooding hazard. These regions are the southernmost part of Chile, north Western Europe, the Baltic Sea, western Alaska, and southern Southeast Asia. As an example, we note that similar differences exist in the delta of the Uruguay river[28] (catchment size of $365 \times 10^3$ km²), consistent with the fact the present analysis does not provide information on compound flooding in long rivers[27].

**Projections of concurrent extremes in precipitation and meteorological tide.** Under a high emissions scenario (RCP8.5), the meteorological drivers of compound flooding are projected to co-occur more frequently along 60% of the global coastline by the end of this century (Fig. 3a). The global median change in return period $\Delta T$ is $-20\%$, which corresponds to joint extreme events becoming 26% more probable. The concurrence probability is projected to increase the most in the Northern Hemisphere at latitudes above 40° north (see red line in Fig. 3b), with the frequency of joint events by 2100 projected to be on average 2.6 times higher compared to present (median $\Delta T = -61\%$). The rise in frequency is particularly evident for coasts in northern North America, northern Europe, northern Mediterranean, Russia, Japan, the Korean peninsula, China, Bangladesh, and around

Cameroon (Fig. 3a). Similar trends are also projected in parts of the Southern Hemisphere, such as for coastlines in northwestern South America, southern Chile, northern Australia, the Gulf of Carpentaria and New Zealand. Concurrent events are projected to become significantly less probable along a smaller portion of the global coastline (see blue line in Fig. 3b), most notably in northwestern Africa, southern Spain, western South Africa, eastern Madagascar, southwestern Australia, and Central Chile. Results are more uncertain in several areas of the tropics and subtropics (magenta line in Fig. 3b), such as in the Caribbean and in Southeastern Asia (see magenta points in Fig. 3a). Overall, note that a large relative decrease in the return period does not necessarily imply a low return period in the future if the present-day return period is relatively high.

**Drivers of changes in concurrent meteorological extremes in coastal areas.** Changes in precipitation extremes are the key driver of the projected changes in compound meteorological extremes (Fig. 4a, b; global median $\Delta T_{\text{prec.}} = -25\%$), increasing the concurrence frequency for 83% of coasts worldwide. Aggregated at global level, the relative contribution of precipitation to changes in concurrence frequency is 77% (Supplementary Table 1). Changes in meteorological tides have a weaker effect (Fig. 4c, d; global median $\Delta T_{\text{met. tide}} = +7\%$). While they reduce the probability of concurrent extremes along 61% of the global coastline, their global relative contribution to the projected changes in concurrence frequency is 20% (Supplementary

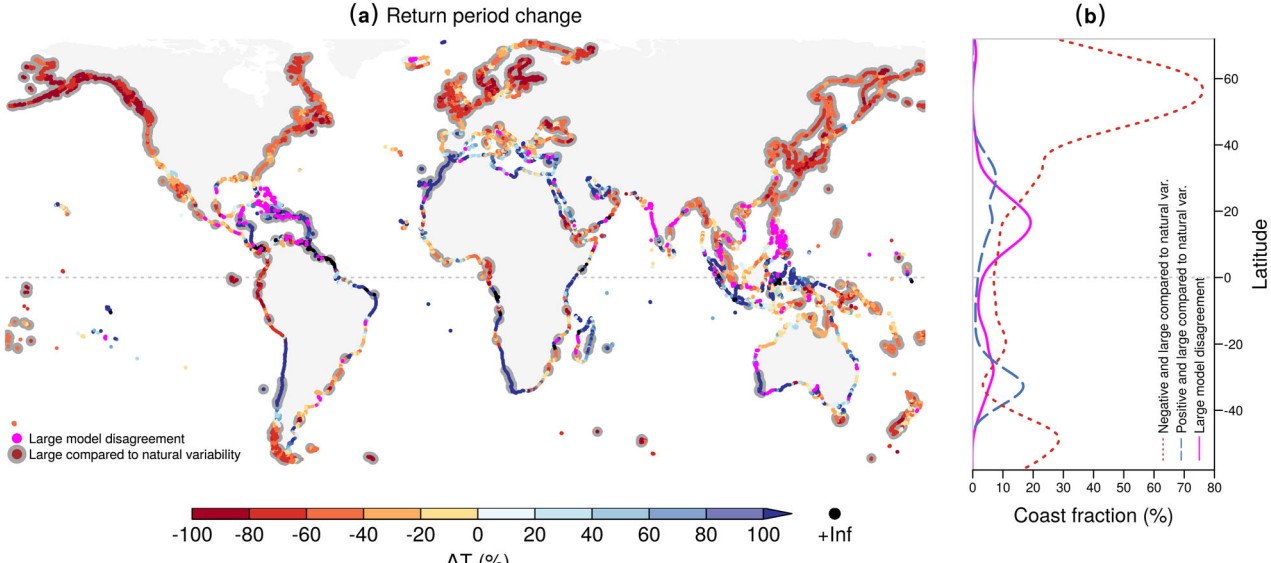

**Fig. 3 Future changes in the return periods of concurrent meteorological drivers of compound flooding. a** Ensemble median projected change (%) of joint return period (or inverse probability) between future (2070–2099) and baseline (1970–2004) climate. Dots with a grey background indicate locations where the projected change is robust, i.e. the ensemble median change lies outside the present-day 95% confidence interval and at least five out of six models agree on the sign of the change. Magenta indicates locations with high model disagreement, i.e. where at least two models project large (lying outside the present-day 95% confidence interval) positive trends and at least two models project large negative trends. **b** Coastline fraction per 5° of latitude (smoothing spline) with a robust negative change (red dotted line), robust positive change (blue dashed line), and high model disagreement (magenta solid line).

Table 1). The effects of changes in the dependence between high meteorological tides and extreme precipitation can be locally pronounced for single climate realisations but the average effects over the ensemble of climate projections are generally small compared to natural variability (Fig. 4e, f). Changes in the dependence structure do not exhibit a clear large-scale spatial pattern and compensate each other at the global scale ($\Delta T_{\text{dep.}} = -1\%$), with approximately balanced fractions of coast with either increasing or decreasing dependency. As a consequence, they increase concurrence probability in 51% of the global coastline, while reducing it in 49%, resulting in an overall global relative contribution of 3% (Supplementary Table 1).

At the regional level, changes in precipitation are the main driver in all IPCC regions apart from South Europe and South Africa where changes in meteorological tides dominate, and the Amazon, North Australia, North East Brazil, Sahara, and West Asia where changes in the dependence dominate the signal (Supplementary Table 1). The dynamics in extreme precipitation increase the probability of concurrent events everywhere apart from the Sahara region. In contrast, projected changes in meteorological tides decrease concurrence frequency everywhere but in Alaska/NW Canada, Central Europe, East and West Africa, East Asia, North Europe, South East South America, and West North America. We move on to examine these regional changes indirectly via discussing the physical processes shaping the large-scale changes in precipitation and meteorological tides, and how they are linked to each other. Changes in the dependences will be discussed in the next section relative to the uncertainty in the projections.

Precipitation extremes are expected to intensify and happen more frequently in most coastal areas worldwide due to the thermodynamic increase in atmospheric moisture content[5], and consequently also the likelihood to have joint inland and coastal meteorological extremes. However, changes in atmospheric circulation can further modulate and potentially oppose this thermodynamic effect. This is the case along the coasts of northern and southwestern Africa, Central Chile, and southwestern Australia, where a more stable atmosphere leads to weaker vertical motion. This, in turn, reduces the intensity of precipitation extremes[5] and consequently the probability for concurrent events in these confined areas (Fig. 4a).

Although the effect of changes in meteorological tide on the projected joint occurrence probability is in general small compared to the natural variability (see Fig. 4d and rare grey dots in Fig. 4c), the spatial patterns are consistent with the projected anthropogenic-driven changes in cyclone activity[44–46]. For example, in Europe, our projections of concurrence frequency are mostly in agreement with the projected decrease in winter storm frequency and intensity in the Mediterranean Sea[45] and the increase in storm track activity in northern Europe[44,45]. This results in a respective decrease and increase in meteorological tides in these two regions[1] and the consequent changes in the concurrence probability. In the Baltic Sea this effect of the increase in meteorological tides is most notable and robust compared to natural variability (grey dots in Fig. 4c). The projected increase in joint occurrence probability for western Canada (Fig. 4c) is consistent with a poleward shift in the north Pacific boreal winter jet stream and in the associated storm-track[44,46,47]. A similar poleward shift of the storm track[46,48] appears consistent with the projected changes in joint return period for New Zealand, southern Australia, and southern South America (Fig. 4c). Lower meteorological tides projected in parts of the equatorial region lead to a reduction in the concurrence probability, especially in Asia (Fig. 4c, d).

Atmospheric circulation dynamics can affect meteorological tide and precipitation extremes in a similar manner, for example through a weakening (or strengthening) of the regional cyclone activity. As a result, meteorological tide-related and precipitation-related changes in the concurrence probability qualitatively demonstrate a positive relation, which is modulated by an offset due to the approximately spatially homogeneous thermodynamic-driven enhancement of precipitation extremes[5] (Supplementary Fig.

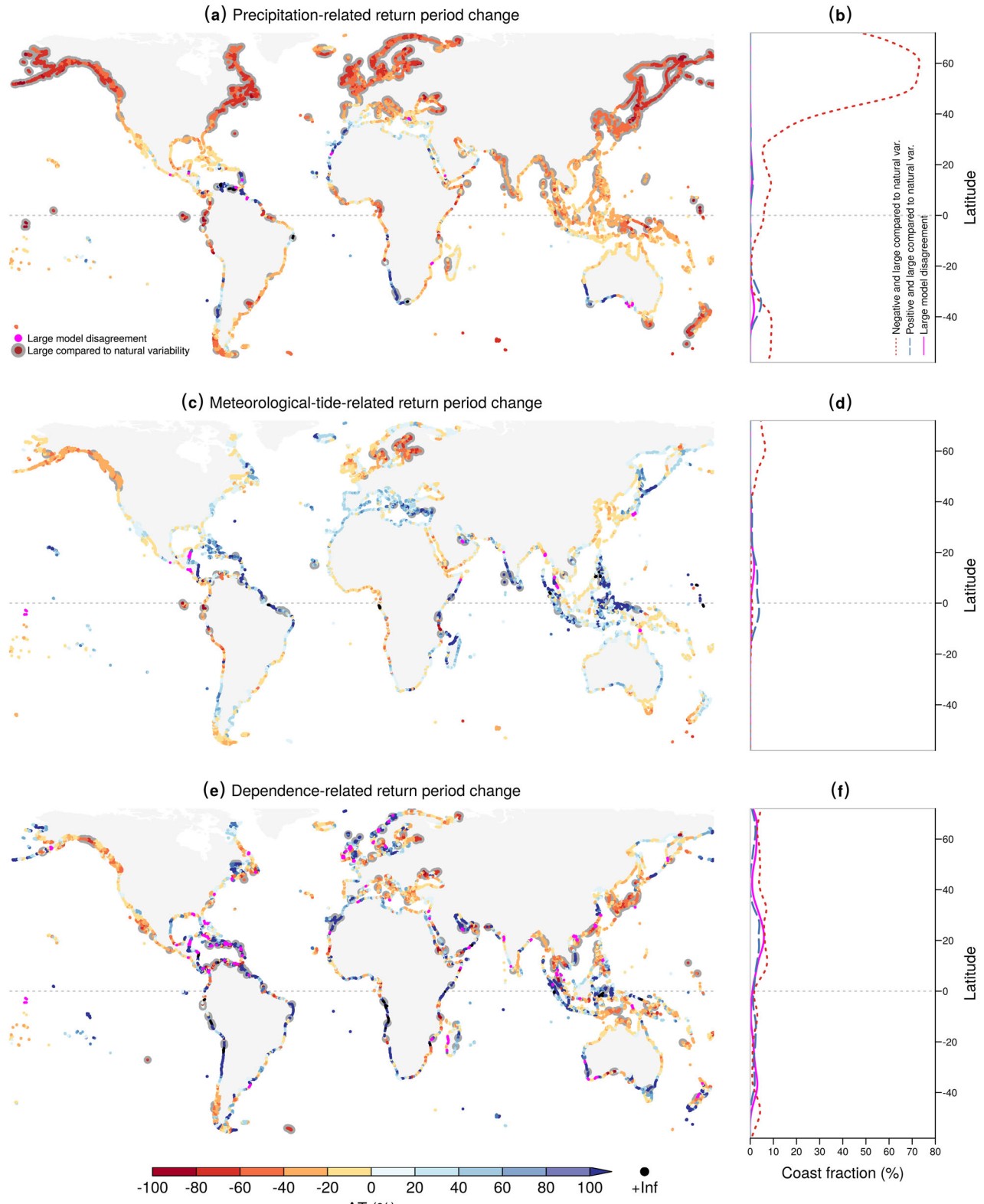

**Fig. 4 Attribution of projected changes in joint return periods to changes in precipitation, meteorological tides, and their dependence. a** Ensemble median projected change (%) of joint return periods (or inverse probability) between future (2070–2099) and baseline (1970–2004) when only taking into account the projected changes in precipitation ($\Delta T_{prec}$). **b** Coastline fraction per 5º of latitude (smoothing spline) with a robust negative change (red dotted line), robust positive change (blue dashed line), and high model disagreement (magenta solid line). **c-f** Similar results as **a** and **b**, but for joint return period changes when only taking into account the projected changes in the meteorological tide ($\Delta T_{met.\ tide}$) in **c-d**, and in the dependence between meteorological tide and precipitation ($\Delta T_{dep.}$) in **e-f**.

S5). For example, reduced cyclonic activity causes a reduction in meteorological tide and precipitation extremes, and therefore in concurrence probability, in northwestern and southern Africa, Central America, Central Chile, and along a large part of Australia's coastlines (compare Fig. 4a and c; see also Fig. 4b from Pfahl et al.[5]). On the contrary, a concurrent increase in meteorological tide and precipitation extremes results in more frequent compound extremes, e.g. in northern Europe, western Canada, and Alaska. The thermodynamic-driven offset is visible, e.g., in the Mediterranean Sea where a weakened cyclonic activity[45] reduces meteorological tides (Fig. 4c) and negatively modulates precipitation[5], but a moistened atmosphere causes more intense precipitation extremes[5] (Fig. 4a; Supplementary Table 1).

**Relative contributions of the drivers to the uncertainty in the projections**. The uncertainty in the projections of future concurrence probability is dominated by the uncertainty in the dependence between meteorological extremes. At the global scale, climate variability in the projected dependence accounts for approximately half (55%) of the uncertainty in the projections of compound meteorological extremes. Precipitation and meteorological tide related uncertainty in the projections are comparable (24% and 21%, respectively) and account for the other half (Fig. 5). Also at the level of IPCC regions, uncertainty in the dependence is the main driver of the future concurrence uncertainty. In certain regions, pronounced but contradicting projections of changes in the dependency (magenta points in Fig. 4e) result in a high uncertainty in the overall response of the concurrence probabilities (magenta points in Fig. 3a), especially in central America which is often hit by TCs. Wahl et al.[8] found that without trends in the records of the individual meteorological drivers, compound events have already increased along some parts of the United States coastline due to a shift towards storm surge weather patterns that also favour high precipitation. The large variability in our projections of the dependence dynamics indicate that there is large uncertainty in how climate change could alter the concurrence of meteorological extremes in addition to the effects of changes in the drivers themselves; large ensemble model simulations would be helpful to disentangle any anthropogenic-driven change in the dependence from natural variability, especially at regional and decadal scales[49]. The findings also indicate that considering the variability of the dependence is crucial to avoid overconfident and potentially misleading projections of future risk.

Uncertainty in the projections of precipitation and meteorological tide extremes is also relevant. Thermodynamic-driven changes in precipitation extremes are a robust feature of climate models. There is, however, less confidence in the magnitude of climate-induced atmospheric circulation changes[50], which exerts a strong control on meteorological tides and regionally modulates precipitation extremes. Hence, despite the consistency among climate models in the sign of the projected changes in precipitation extremes, the magnitude of the changes is uncertain, especially in the tropics[5]. At midlatitudes, projections of wind extremes that modulate meteorological tides also show large uncertainty as a result of uncertainty in the evolution of atmospheric circulation[47].

## Discussion

This study provides the first global assessment of the effects of climate change on the meteorological drivers of compound flooding. Our estimates should not be interpreted as actual compound flood hazard[10,24,26,31]. Rather, we studied the probability of co-occurrence of extremes in meteorological tide and in precipitation in coastal areas, which provides insight into large-scale rainfall-driven compound flooding in low-lying coasts and compound flooding in estuaries of small-size and medium-size rivers[27]. When meteorological drivers of compound flooding co-occur, the actual flooding will depend on a variety of additional factors. For example, coastal flooding usually happens during high tides[10,51] that do not depend on meteorological conditions. In addition, actual flooding only takes place when flood protection or natural barriers are overtopped or breach and low-lying lands become inundated.

By the end of this century, SLR could push up mean sea levels by one meter or more[1]. This upward shift will strongly increase the probability to experience what is today an extreme meteorological tide and consequently of compound flooding if coastal protection is not adjusted for SLR[29]. We show that changes in the joint probability of high meteorological tides (on top of the

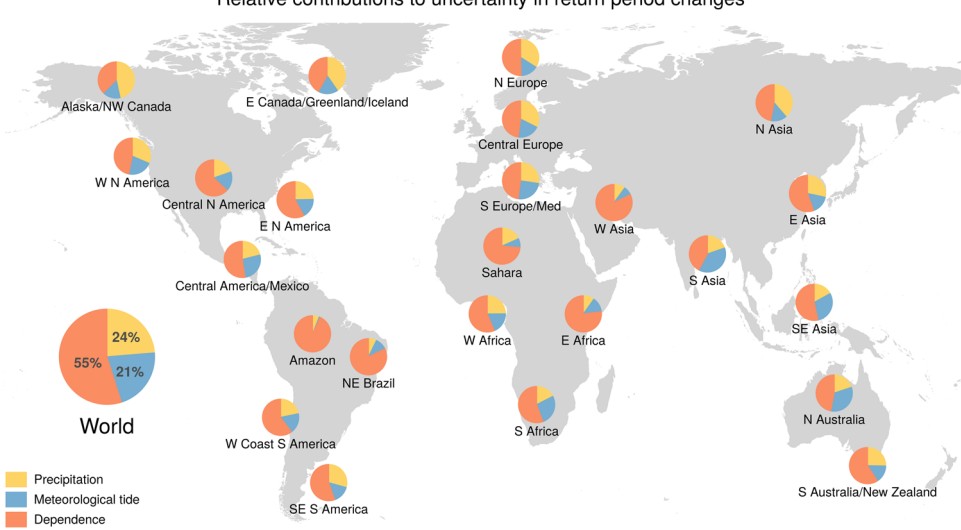

Relative contributions to uncertainty in return period changes

**Fig. 5 Drivers of uncertainty in the projected changes of the joint return period of concurrent precipitation and meteorological tide extremes for IPCC subregions and worldwide.** Relative model uncertainty in the projected change in joint return periods (or inverse probability) between the future (2070–2099) and baseline (1970–2004) climates driven by the individual meteorological drivers of compound flooding, i.e. only by precipitation, meteorological-tide, and their dependence (see "Methods" section). The IPCC subregions are shown in Supplementary Fig. S6.

elevated mean sea level) and extreme precipitation could also considerably affect compound flooding along coastlines worldwide. Our projections vary greatly between regions, but globally averaged the triggering conditions of compound flooding are projected to become one-fourth more probable under a high emissions scenario by 2100. At latitudes above 40º north the factor rises to more than 2.5 times. Despite the limitations inherent to a global scale analysis[10,25,26,37,52] (see also discussion in a dedicated section in the "Methods" section), our study provides insight into the large-scale spatio-temporal dynamics of compound flood drivers in the context of climate change. Local assessments focussing on hotspot regions with robust projected changes could provide more accurate assessments of compound flood hazard and its impacts[12,13,18,53].

Coasts will be particularly exposed to the effects of global warming through a range of climate change processes[54]. Adaptation targeted at protecting coastal communities should not only take into account expected SLR[2,55] or projections of total extreme sea levels[56], but also dependencies with inland meteorological extremes. Neglecting compound flood hazard and changes therein might leave several parts of the global coastline insufficiently protected. Concurrent extreme events may be more challenging compared to individual coastal or inland flooding events also for taking emergency response action, e.g. they may amplify impacts through overloading rescue teams[57,58]. The presently reported increasing frequency of such concurrent hazards highlights the need for better planning of emergency response and designing efficient protective structures.

## Methods
**Data.** Meteorological tides, from Vousdoukas et al.[1], were obtained by adding the storm surge level ($\eta_{\text{storm surge}}$) and the wave setup ($H_s$), with the latter approximated as the significant wave height multiplied by 0.2, i.e. as $\eta_{\text{storm surge}} + 0.2H_s$. Waves were simulated with Wavewatch III[1,32,59] (forced with 6-hourly wind field). Storm surges were modelled with D-FLOW Flexible Mesh (FM) using a flexible mesh setup (forced by 6-hourly wind and atmospheric pressure fields)[1,25,59,60]. The resulting sea level data are available every ~100 km along the global coastline. Precipitation was taken from the climate model grid point nearest to each coastal location. Detailed information on the sea-level dataset and models can be found in refs. [1,10,25,32,59,60], including data validation. Our analysis is based on quantile values; therefore, we did not bias correct simulated data[10]. Sea level and precipitation data are based on ERA-Interim[34] (period 1980–2014) and six CMIP5 models[35] (periods 1970–2004 and 2070–2099) (i.e., ACCESS1-0, ACCESS1-3, GFDL-ESM2M, GFDL-ESM2G, CSIRO-Mk3-6-0, and EC-EARTH). The GFDL-ESM2G model is not considered for the Black Sea and the Red Sea coasts because of instabilities of the surge model.

**Discussion about sea level modelling.** The results should be interpreted considering some inevitable limitations, which are common in large-scale studies. As discussed above, higher-resolution input data, such as from new ERA5 reanalysis and HighResMIP, could be used in future work to improve the representation of meteorological tides[1,25]. Especially for TCs, the passage of a tropical cyclone will be associated with a meteorological tide peak that can be underestimated[25,37–39]. However, such an underestimation is likely alleviated by the design of our analysis. In fact, we are not considering absolute storm intensities, but relative values. That is, we define extremes based on percentiles of the grid point distributions of meteorological tides and precipitation, which at least reduces the effect of biases in the magnitudes of the absolute values. We have not considered the locations above the Polar circle that are affected by ice-related processes not properly resolved by our ocean models[1]. Physical interactions between waves, storm surges, astronomical tides, and SLR are not resolved[61–64]. Therefore it is assumed that sea level components are independent. This is inevitable for a global analysis given the current modelling capacity. However, studies have demonstrated that assuming no interaction between the sea level components is acceptable given the overall uncertainty in the climate change projections[1,65–67]. Overall, we are confident that the above limitations do not distort our findings and that our study expands the understanding of present and future global compound flooding hazard. This is supported by the fact that our findings are physically consistent with studies of large-scale atmospheric circulation changes employing larger model ensembles than the present one[5,44,45,46,47,48,68,69].

**Cyclone tracking.** Extratropical cyclone (ETC) tracks were identified based on the objective feature tracking algorithm TRACK[70,71]. Following Hoskins and Hodges

(2002)[72], the algorithm uses the 850 hPa relative vorticity (from ERA-Interim) to identify and track cyclones in both hemispheres. Tropical cyclone tracks were obtained from the observation-based IBTrACS dataset[73]. For both the tropical and ETC datasets, the spatial maps of cyclone track density were computed using spherical kernel density estimators[74].

**Return periods.** We assessed the bivariate return periods[75] of concurring heavy precipitation and high sea-level (individual 99.7th percentiles). The bivariate return period that we used—so-called "AND"[42,76,77]—allows for disentangling flooding caused by the concurrence of high sea-level and precipitation values. To estimate return periods, we applied a parametric copula-based bivariate probability distribution to selected pairs of meteorological component of sea-level (i.e., meteorological tide) and precipitation that are simultaneously high, i.e., that exceed the individual 95th percentiles ($s_{\text{sel}}$ and $p_{\text{sel}}$, respectively). In locations where few pairs were selected we reduced the selection threshold to below 0.95 in order to ensure that at least 20 pairs of values were selected. We replaced groups of selected event pairs separated by <3 days by a unique event having the maximum precipitation $P$ and meteorological component of sea-level $S$ observed in the group. We define the bivariate return period as

$$T(s_{99.7}, p_{99.7}) = \frac{\mu}{P((s>s_{99.7} \text{ and } p>p_{99.7}) \mid (s>s_{\text{sel}} \text{ and } p>p_{\text{sel}}))}$$
$$= \frac{\mu}{1 - u_{S99.7} - u_{P99.7} + C_{SP}(u_{S99.7}, u_{P99.7})}, \quad (1)$$

where $\mu$ is the average time elapsing between the selected pairs, $u_{P99.7} = F_P(p_{99.7})$, $F_P$ is the marginal cumulative distribution of the precipitation variable within the selected pairs (accordingly for sea level), and $C_{SP}$ is the copula modelling the dependence between the selected pairs. We fitted copulas from the families Gaussian, $t$, Clayton, Gumbel, Frank, Joe, BB1, BB6, BB7, BB8 to ($u_S$, $u_P$) (obtained via empirical marginal cumulative distribution function (CDF)[77], and then we selected the best-ranked family according to the Akaike information criterion. We modelled the marginal distributions of precipitation and sea level beyond the selection thresholds by a Generalised Pareto Distribution. Copulas and marginal distributions were fitted through a maximum-likelihood estimator (using the *VineCopula*[78] and *ismev*[79] R-packages). The goodness of fit of copulas and marginals was tested via the Cramer–von-Mises criterion (via the *VineCopula*[78] and *eva*[80] R-packages, respectively).

**Return period (and probability) changes and robust changes.** For the individual CMIP5 models, the changes (%) in the return periods (e.g., in Fig. 3a) were estimated as $\Delta T(\%) = 100 \times (T^{2070-2099} - T^{1970-2004})/T^{1970-2004}$. Based on physical interpretation, when both $T^{2070-2099}$ and $T^{1970-2004}$ are infinite, $\Delta T(\%)$ is set to 0%; when $T^{2070-2099}$ is finite and $T^{1970-2004}$ is infinite, $\Delta T(\%)$ is set to −100%. A *robust* (large compared to natural variability) return period change is defined as the case where the multi-model median of the return period change lies outside the present-day 95% range due to natural variability (estimated with a resampling approach[8], see the next section) and at least five out of six models agree on the sign of the change.

Given the inverse relationship between return period and probability[42], it can be estimated that $\Delta P(\%) = -100 \cdot \Delta T(\%)/(\Delta T(\%) + 100)$, where the percentage change in probability is defined similarly to the percentage change in return period.

**Present-day range of the return period due to natural variability.** The present-day range in the return period due to natural variability was estimated (for ERA-Interim) as a 95% confidence interval based on resampling the interannual variability. For each location, we randomly sampled $N_{\text{bootstrap}} = 700$ bivariate time series of precipitation and meteorological tides, and computed the associated 700 return periods. Each of these 700 time series has the same length as the original time series, and was built through combining randomly sampled calendar years of the precipitation and meteorological tide bivariate time series. This procedure is preferred to a classic resampling of the daily pairs, as it allows for preserving the autocorrelation of the variables. The final 95% confidence interval was then defined as the 2.5–97.5th percentile interval of these 700 return periods, i.e. ($T_{2.5\text{th}}$, $T_{97.5\text{th}}$). The associated 95% confidence interval of the variations in the return period due to natural variability was computed as the percentage difference between the observed return period and ($T_{2.5\text{th}}$, $T_{97.5\text{th}}$).

**Partitioning of return period changes.** In Fig. 4, we show the assessment of how the return periods would change in the future when only taking into account changes (with respect to the present) of (1) the precipitation marginal distribution (i.e. the full distribution of the precipitation without reference to meteorological tide), (2) the meteorological tide marginal distribution, and (3) the dependence between the precipitation and meteorological tides[10,13,81]. We computed the change in the return period (%) for the case (i) as $\Delta T_{\text{exp i}}(\%) = 100 \cdot (T^{\text{fut}}_{\text{exp i}} - T^{\text{pres}})/T^{\text{pres}}$, where $T^{\text{pres}}$ is the return period for the present period and $T^{\text{fut}}_{\text{exp i}}$ is computed as follows. Case (1): we get the empirical cumulative distribution $U_{S_{\text{pres}}}$ of the present-day sea level $S_{\text{pres}}$ as $U_{S_{\text{pres}}} = F_{S_{\text{pres}}}(S_{\text{pres}})$ (where $F_{S_{\text{pres}}}$ is the empirical CDF of $S_{\text{pres}}$). We define the empirical

CDF $F_{S_{\text{fut}}}$ of the variable $S_{\text{fut}}$, and define $S_1 = F_{S_{\text{fut}}}^{-1}(U_{S_{\text{pres}}})$. As a result, the variables $(S_1, P_{\text{pres}})$, where $P_{\text{pres}}$ is the present-day precipitation, have the same dependence (Spearman correlation and tail dependence[13]) as during the present, but the marginal distribution of $S_1$ is that of the future. The return period $T_{\text{exp 1}}^{\text{fut}}$ was computed based on $(S_1, P_{\text{pres}})$. Case (2) was obtained as case (1), but switching precipitation and sea level variables in the procedure above. Case (3): we define $S_3 = F_{S_{\text{pres}}}^{-1}(U_{S_{\text{fut}}})$, where $U_{S_{\text{fut}}}$ is the empirical cumulative distribution of the future sea level and $F_{S_{\text{pres}}}$ is the empirical CDF of the sea level in the present climate. Similarly, we get $P_3 = F_{P_{\text{pres}}}^{-1}(U_{P_{\text{fut}}})$. As a result, the variables $(S_3, P_3)$ have the same dependence as during the future, but the marginal distributions of the present climate[13]. Then, the return period $T_{\text{exp 3}}^{\text{fut}}$ was computed based on $(S_3, P_3)$. We observe that the total change in the return periods is not given by the sum of the changes estimated in these three cases above (the return period is not given by a linear combination of the overall marginal distribution and the dependencies).

**Partitioning of the uncertainties in the return period changes**. In Fig. 5, we quantify the relative importance of the uncertainty in the projected changes of the three meteorological drivers (precipitation, meteorological tides, and their dependence) for the uncertainty in the return period changes. This is achieved through first computing, for the three drivers above, the symmetrized changes of the return periods for the six CMIP5 models, which are defined in the next section and are referred to as $\Delta T_{\text{exp}}^{\text{Symmetric}}$ (%) from now on.

Then, three steps are necessary, for each of the three meteorological drivers (i):

1. Quantify the uncertainty in the return period change due to changes in the driver (i) via the intermodel spread of the $\Delta T_{\text{exp i}}^{\text{Symmetric}}$ (%), from now on referred to as $\sigma(\Delta T_{\text{exp i}}^{\text{Symmetric}}$ (%)). Note that the intermodel spread is defined as the difference between the second highest and second lowest among the changes projected by the six climate models.
2. For a given IPCC region $r$ (shown in Supplementary Fig. S6), compute the regional median of $\sigma(\Delta T_{\text{exp i}}^{\text{Symmetric}}$ (%)), from now on referred to as $\sigma_{i,r}$.
3. Finally, quantify the regional relative importance of the driver (i) for the uncertainty in the return period changes as $100 \cdot \sigma_{i,r}/(\sum_{i=1}^{3} \sigma_{i,r})$.

**Symmetrized changes of the return periods for computing uncertainties in return period future changes**. For a generic return period, given the percentage change of the return period $\Delta T(\%)(T^{\text{fut}}, T^{\text{pres}}) = 100 \times (T^{\text{fut}} - T^{\text{pres}})/T^{\text{pres}}$, the symmetrized change of the return period is defined as

$$\Delta T^{\text{Symmetric}}(\%)(T^{\text{fut}}, T^{\text{pres}}) = \begin{cases} \Delta T(\%)(T^{\text{fut}}, T^{\text{pres}}) & \text{if } \Delta T(\%)(T^{\text{fut}}, T^{\text{pres}}) > 0 \\ -\Delta T(\%)(T^{\text{pres}}, T^{\text{fut}}) & \text{otherwise,} \end{cases}$$

(2)

where $-\Delta T(\%)(T^{\text{pres}}, T^{\text{fut}}) = -100 \times (T^{\text{pres}} - T^{\text{fut}})/T^{\text{fut}}$.

In the following, we provide an explanation of why we use these symmetrized changes. $\Delta T^{\text{Symmetric}}(\%)$ are preferred to simple changes $\Delta T(\%)$ as the latter tends to skew the magnitude of the uncertainty for negative return period changes. In fact, $\Delta T(\%)$ assumes values between $-100$ and $0$ and between $0$ and $+\text{Inf}$ for negative and positive changes of the return periods, respectively. As a result, the uncertainty (intermodel spread) of $\Delta T(\%)$ would tend to appear smaller where models show a reduction of the return periods, and larger where models show an increase of the return periods. $\Delta T^{\text{Symmetric}}(\%)$ avoids this issue as it is defined such that:

- It assumes values between $-\text{Inf}$ and $0$ and between $0$ and $+\text{Inf}$ for negative and positive changes of the return periods, respectively. This implies that, e.g., a doubling and halving of $T^{\text{fut}}$ (with respect to $T^{\text{pres}}$) corresponds to equal but opposite values of $\Delta T^{\text{Symmetric}}(\%)$, which is not true for $\Delta T(\%)$.
- It has the desirable property of detecting the same return periods' intermodel spread in, e.g., four situations where the six CMIP5 models project $T^{\text{fut}} = (1, 2, 3, 4, 5, 6) \cdot T^{\text{pres}}$; $T^{\text{fut}} = (1, 1/2, 1/3, 1/4, 1/5, 1/6) \cdot T^{\text{pres}}$; $T^{\text{fut}} = (1/3, 1/2, 1, 2, 3, 4) T^{\text{pres}}$; and $T^{\text{fut}} = (11, 12, 13, 14, 15, 16) T^{\text{pres}}$. (This occurs because $\Delta T^{\text{Symmetric}}(\%)$ increases linearly with $a$, where $T^{\text{fut}} = a \cdot T^{\text{pres}}$, and decreases linearly with $b$, where $T^{\text{fut}} = 1/b \cdot T^{\text{pres}}$.)

## Data availability
Precipitation data from ERA-Interim are available from the ECMWF Public Datasets web interface (http://apps.ecmwf.int/datasets). Precipitation data from CMIP5 models are available from the Earth System Grid Federation (ESGF) Peer-to-Peer system (https://esgf-node.llnl.gov/projects/cmip5). Sea level data are available at https://data.jrc.ec.europa.eu/collection/liscoast (further inquiries should be addressed to M.I.V.).

## Code availability
The statistical analyses were carried out using the R packages cited in Materials and Methods. Custom codes developed for the analyses are available from the corresponding author upon reasonable request.

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

## Acknowledgements

E.B. and G.Z. acknowledge financial support from the European Research Council grant ACRCC (project 339390). E.B. acknowledges support from the Volkswagen Foundation's (CE:LLO project, grant no. 88468). E.B. and D.M. acknowledge the European COST Action DAMOCLES (CA17109).

## Author contributions

E.B. conceived the study, carried out the data analysis, and drafted the manuscript. E.B. designed the study development with contributions from G.Z. and M.I.V. E.B., M.I.V., L.F., and T.G.S. worked on the final manuscript with contributions from G.Z. M.I.V. and L.M. performed the storm surge and wave runs. K.H. analysed the storm tracks. All authors (E.B., M.I.V., G.Z., K.H., T.G.S., D.M., L.M., and L.F.) discussed the results and commented on the manuscript.

## Competing interests

The authors declare no competing interests.

## Additional information

**Peer review information** Primary handling editor: Heike Langenberg

