## [Peer Review File · Communications Earth & Environment]

Web links to the author's journal account have been redacted from the decision letters as indicated to maintain confidentiality.

16th Jun 20

Dear Dr Bevacqua,

Your manuscript titled "Global projections of compound coastal meteorological extremes" has now been seen by two reviewers, whose comments are appended below. You will see that they find your work of some potential interest. However, they have raised quite substantial concerns that must be addressed. In light of these comments, we cannot accept the manuscript for publication, but would be interested in considering a revised version that fully addresses these serious concerns.

We hope you will find the reviewers' comments useful as you decide how to proceed. Should additional work allow you to

- address these criticisms (that is, either to incorporate the suggestions or provide a compelling argument why the point made by the reviewer is not valid, or relevant to the editorial threshold as outlined below)

AND

- meet our editorial thresholds as outlined below,

then we would be happy to look at a substantially revised manuscript.

In the following, we list our main editorial concerns that we consider most relevant to these threshold points.

***Editorial threshold 1: Present a compelling quantification of present and future risk from compound events of coastal flooding from river flooding combined with storm surges/meteorological tides

***Editorial threshold 2: Clarify the advance the present paper represents over the study by Eilander et al (2020), noted by reviewer 2 (<https://iopscience.iop.org/article/10.1088/1748-9326/ab8ca6>), including a full discussion that compares and contrasts the results from both papers.

***Editorial threshold 3: Improve the accessibility and transparency of the manuscript by clearly laying out the research question in the beginning, defining all key concepts and terms early on (e.g. "meteorological tide"), and providing sufficient detail on the methods to make the study reproducible.

However, please bear in mind that we will be reluctant to approach the reviewers again in the absence of substantial revisions.

If the revision process takes significantly longer than three months, we will be happy to reconsider your paper at a later date, as long as nothing similar has been accepted for publication at Communications Earth & Environment or published elsewhere in the meantime.

We understand that due to the current global situation, the time required for revision may be longer than usual. We would appreciate it if you could keep us informed about an estimated timescale for resubmission, to facilitate our planning. Of course, if you are unable to estimate, we are happy to accommodate necessary extensions nevertheless.

Please use the following link to submit your revised manuscript, point-by-point response to the referees' comments (which should be in a separate document to any cover letter) and any completed checklist:

[link redacted]

Please do not hesitate to contact me if you have any questions or would like to discuss the required revisions further. Thank you for the opportunity to review your work.

Best regards,

Heike Langenberg, PhD

Chief Editor
Communications Earth and Environment

On Twitter: @CommsEarth

EDITORIAL POLICIES AND FORMAT

If you decide to resubmit your paper, please ensure that your manuscript complies with our editorial policies and complete and upload the checklist below as a Related Manuscript file type with the revised article:

Editorial Policy Policy requirements

For your information, you can find some guidance regarding format requirements summarized on the following checklist:

Communications Earth & Environment formatting checklist

REVIEWER COMMENTS:

Reviewer #1 (Remarks to the Author):

Communications Earth & Environment manuscript

Manuscript number: COMMSENV-20-0169-T

Title: Global projections of compound coastal meteorological extreme

Compound climate or weather events are a pressing topic in recent years. So far, many scholars have been focusing on this field and a bunch of papers have published. Yet, most compound studies focus on the extreme climate extreme. This paper eyes on the compound coastal and inland floods, which delivers some novel pattern and findings over global coastal areas. Also the future patterns and variability in the compound extreme meteorological tides and precipitation were examined by using the copula as a baseline method. The relative contribution of climate and tides as well as the dependence in the concurrence probability over a few global sub-areas were identified. The general impression about this paper is already well-written and structured. I would recommend a moderate revision before it could be published. Yet, I do have some concerns as follows:

1. The introduction reads like not clear enough to see why the author need to explore such studies? Simply because some ones have done similar studies? I would suggest more in-depth explanations about the basic hypothesis and their motivation are expected.

2. Lines 37-45, these lines seem focusing on more about how the method is developed or applied. One may expect to see some points about the main aspects that the paper tries to address, instead of how it is developed.

3. An important question about the method. When you compare the difference in concurrence probability between the present and future, which period was used as reference period (climatology). This defined reference period may strongly affect the variability in concurrence probability for both preset and future periods.

4. Also related to the method, both subsections "Symmetrized changes of the return periods for computing uncertainties in return period future changes" and "Partitioning of return period changes and associated relative uncertainties" are not easy to follow for the readers not familiar with the joint probability. It also takes time for me to understand what the authors are analyzing. I would suggest a clear step-by-step procedure for both sections is needed.

5. Figure 3, the Present-day return periods of concurrent extremes are interesting. It seems the concurrence probability in Northern hemisphere is much higher than the southern part. Do the authors have any ideas about the potential reasons? And the equatorial coastals have much low concurrence probability, while our general impression is that such areas have more extreme rainfall.

6. It is not clear enough how the length of the concurrent extreme season was calculated. Does that figure mean that somewhere the concurrent extremes could last for 9 months?

7. Another major issue is about the extreme thresholds defined. The current threshold is 0.95 or 0.997. Is that too "extreme"? In that case, there are not too many cases left when you selected the extreme events, particular for the compound extreme. Why not define 75% or 80% quantiles, probably not that "extreme", but may cover more events which may significantly reduce the uncertainty caused by the scarce available data points.

8. One important result is "latitudes above 40 degree north, compound meteorological extremes would become more than 2.5 times as frequent". I think such large difference sometime means little important information. For instance, based on the current thresholds used, the present concurrence probability may be around 0.0001, and for the future period, that probability is around 0.00025, which also means 2.5 times increased. But such event in fact is also impossible to occur, because the probability is too low. Thus, I would suggest the author consider lower thresholds to define individual

extremes or the authors have some explanations on this issue.

9. The relative contribution of driver for each sub-area is interesting. But there is lack of in-depth analysis or discussion why there are large difference between sub-aeras about the drivers.

Reviewer #2 (Remarks to the Author):

See attached

Nature Communications, peer-review stage 1

MANUSCRIPT: Global projections of compound coastal meteorological extremes

Bevacqua and co-authors have analysed projections of change in co-occurring extremes of coastal precipitation and meteorological tides at the global scale. Such compound events are relevant, because if present, would increase likelihood of extreme floods, with obvious consequences for coastal communities.

Based on reanalysis data they show that such concurrent extremes are widespread, with a global average return period of 18 years. The two events are thus frequently linked, this return period is 20 times higher than would be expected for truly independent extremes. Using six CMIP5 models and a high-end scenario (RCP8.5) they then go on to show robust increases for many coasts, especially polewards of 40 N/S. These changes are dominated by changes in precipitation extremes, and in some regions a smaller change in meteorological tides or dependence.

The topic of research is of interest, but I wonder if issues with TCs, or the lack of river/land processes have been sufficiently dealt with to warrant publication in this high-impact journal. In its current form, the manuscript paints too big a picture in the introduction. The results do not fill the knowledge gap painted. Hence, I recommend major revisions, either by modifying the analysis or writing a more fitting introductory text, before publication of this manuscript.

MAJOR

- In the introduction the importance of compound flooding is noted, and add three TC-based examples (Harvey, Irma, Maria) are given. The fact that specifically TCs are unsatisfactorily simulated in CMIP models is a problem (which you note in the method section only). Though I realise the difficulty, I think this is something that needs to be solved before publication of the manuscript. One solution is to use HighResMIP data instead of CMIP data, at 25-50 km scale coupled models start to simulate hurricane force winds and more realistic TC spatial patterns. If this is unfeasible given computation costs, note early on that TCs are not included reliably in the investigation.
- In your abstract and introduction you talk about compound flooding and its importance. The paper then discusses the joint probability of extremes in meteorological tide and precipitation. These are not the same thing, as you note for the first time in the discussion (line 159). Please add these important details at the start of the paper too, overselling of what you are studying does not help the manuscript. A reference to Eilander et al. (2020, doi 10.1088/1748-9326/ab8ca6) is warranted as well, they provide a global analysis of compound flooding and do take into account riverine processes.
- Why use an incomplete ERA-Interim dataset (1980-2014, not 1979-2019)? This should be updated. Better would be to use ERA5, the improved state-of-the-art reanalysis product from ECMWF.

MINOR

- Lines 39-41: This sentence is not clear, especially "forced with reanalysis data for the present and with CMIP5 GCM climate projections up to...". Maybe rewrite along the lines "forced with reanalysis data for the observed past, and with CMIP5 GCM projections for estimates of future climates."
- Line 60: I'd note cyclones normally travel away from the coast near Central America.
- Line 71: Why change the name to 'inland extremes'?
- Line 182: Please add why emergency response is different for floods by only meteorological tide, or floods due to both. I doubt response is hampered because of rain.
- Figure 1: Why are large parts of northern Russian coast/Canada/Greenland not modelled?
- Figures: Given that we are talking about wet extremes (rain and tide), I suggest swapping you color scales to show high co-occurrence in blue.
- Figure 2c: Does colouring by season rather than calendar month provide more information? Large differences between northern and southern hemisphere are not a dynamical feature, but result of course from differences in solar insolation.
- Figure 5: Can you scale the circles by the absolute uncertainty in projected changes, such that large circles show larger uncertainties? If absolute uncertainty is low, the source of that uncertainty is less relevant.

Response to the reviewers

We would like to thank the reviewers and the editor for the time spent in reviewing the paper. We found the comments and suggestions to be very valuable and constructive. We firmly believe that they have contributed to substantially improving the manuscript.

Please, note that in the new version of the manuscript, we have decided not to consider the coastal locations above the Polar circle that are typically exposed to sea-ice in winter. As explained in the original version of the manuscript, results in these locations need to be considered with caution given that state-of-the-art ocean models do not consider the interaction between sea ice and meteorological tides (e.g., Hemer et al., 2013; Vousdoukas et al., 2018). This change has led to a few minor changes in the presentation of the results. However, the overall results and conclusions of the paper are not affected at all.

Please find the response to the individual comments from the reviewers below. A new version of the manuscript, including changes based also on the comments from the editor, is attached.

Best regards,

Emanuele Bevacqua, Michalis I. Vousdoukas, Giuseppe Zappa, Kevin Hodges, Theodore G. Shepherd, Douglas Maraun, Lorenzo Mentaschi, and Luc Feyen.

Hemer MA, Fan Y, Mori N, Semedo A, Wang XL. Projected changes in wave climate from a multi-model ensemble. *Nature climate change*. 2013 May;3(5):471-6.

Vousdoukas MI, Mentaschi L, Voukouvalas E, Verlaan M, Jevrejeva S, Jackson LP, Feyen L. Global probabilistic projections of extreme sea levels show intensification of coastal flood hazard. *Nature communications*. 2018 Jun 18;9(1):1-2.

Reviewer #1 (Remarks to the Author)

Compound climate or weather events are a pressing topic in recent years. So far, many scholars have been focusing on this field and a bunch of papers have published. Yet, most compound studies focus on the extreme climate extreme. This paper eyes on the compound coastal and inland floods, which delivers some novel pattern and findings over global coastal areas. Also the future patterns and variability in the compound extreme meteorological tides and precipitation were examined by using the copula as a baseline method. The relative contribution of climate and tides as well as the dependence in the concurrence probability over a few global sub-areas were identified. The general impression about this paper is already well-written and structured. I would recommend a moderate revision before it could be published. Yet, I do have some concerns as follows:

We would like to thank the reviewer for the positive feedback on our work and highlighting that it delivers novel patterns and findings over global coastal areas.

1. The introduction reads like not clear enough to see why the author need to explore such studies? Simply because some ones have done similar studies? I would

suggest more in-depth explanations about the basic hypothesis and their motivation are expected.

We have expanded the current discussion in the introduction. In particular we have highlighted the motivation for and relevance of performing such a study. We write:

“In the future, sea level rise (SLR) resulting, e.g., from thermal expansion and melting of continental glaciers and polar ice sheets, will push mean and extreme sea levels upward[1] and will thereby increase the future compound flood hazard[29, 10]. However, meteorological drivers of compound flooding such as extreme precipitation, meteorological tide, and their interplay will also be affected by climate change[5, 30, 1]. For example, a warmer atmosphere will favour an increase in the atmospheric moisture content, resulting in more intense precipitation extremes in most coastal areas worldwide[5, 30]. Changes in storm frequency and intensity will affect meteorological tides, and are expected to result in associated changes in extreme sea level[1]. Therefore, it is likely that the potential for compound flooding will change along with the changes in these driving meteorological processes, beyond the effects driven by mean SLR. This has been shown for Europe’s coasts[10], but such information is currently missing for most low-lying coastal areas around the world. The above, in combination with the expected future increase in coastal population, highlights the need for a comprehensive assessment of the meteorological drivers of compound flooding and their response to climate change.”

2. Lines 37-45, these lines seem focusing on more about how the method is developed or applied. One may expect to see some points about the main aspects that the paper tries to address, instead of how it is developed.

Consistent with the guidelines of the journal, the Methods section is separated from the main body of the manuscript, therefore we wish to provide a brief presentation of the main methodological aspects in the main text, such that the paper may be followed by the reader prior to checking technical details in the Method section. This is because the methodology is key to the nature of the evidence we provide. We nevertheless agree with the referee that expanding the discussion of the main aspects addressed in the paper would be helpful to the reader at this point of the introduction. Therefore, prior to engaging in the discussion of the methodological aspects (which have been moved after the introduction section in the revised manuscript), we have added a paragraph, where we state:

“We first assess the present-day (1980-2014) probabilities of concurring meteorological extremes, including an analysis of their seasonality and physical drivers through focusing on storm tracks. Second, we analyse the changes in the compound extremes by the end of the century (2070-2099) compared to the recent past (1970-2004), and highlight areas with the largest trends. Third, we disentangle, quantify, and interpret the contribution of the meteorological drivers of compound flooding as well as the dependence between them[10, 8] to the overall change. Finally, we investigate the uncertainties in the changes and how these are related to those of the meteorological drivers.”

3. An important question about the method. When you compare the difference in concurrence probability between the present and future, which period was used as reference period (climatology). This defined reference period may strongly affect the variability in concurrence probability for both preset and future periods.

We apologize for not being sufficiently clear on this. In fact, in the original manuscript, the periods employed in the study were defined in the figure captions and Methods section only. We agree that, as for all climate change studies, the detected changes and baseline variability of the variable of interest are dependent on the definition of the reference/baseline period. Selecting two fixed time periods is a standard choice in the literature. Here, as done in a previous assessment of the present and future compound flooding in Europe (Bevacqua et al., 2019), we use the years 1970-2004 for the baseline period and 2070-2099 future period. Given the comment of the reviewer, we have defined the periods explicitly also in the main text (we did it as shown in the comment above).

Bevacqua E, Maraun D, Vousdoukas MI, Voukouvalas E, Vrac M, Mentaschi L, Widmann M. Higher probability of compound flooding from precipitation and storm surge in Europe under anthropogenic climate change. *Science advances*. 2019 Sep 1;5(9):eaaw5531.

4. Also related to the method, both subsections “Symmetrized changes of the return periods for computing uncertainties in return period future changes” and “Partitioning of return period changes and associated relative uncertainties” are not easy to follow for the readers not familiar with the joint probability. It also takes time for me to understand what the authors are analyzing. I would suggest a clear step-by-step procedure for both sections is needed.

We have modified the presentation of these sections through reducing the amount of text, presenting them in a different order, and defining a step-by-step procedure as suggested by the referee.

5. Figure 3, the Present-day return periods of concurrent extremes are interesting. It seems the concurrence probability in Northern hemisphere is much higher than the southern part. Do the authors have any ideas about the potential reasons? And the equatorial coasts have much low concurrence probability, while our general impression is that such areas have more extreme rainfall.

Thanks for the very relevant comment, we had not noticed this interesting aspect. In fact, there is a statistically significant difference between the median return period in the Northern Hemisphere (15 years) and Southern Hemisphere (23 years). First, we ensure that differences do not arise by a simple asymmetry in the latitudinal extension of the coastal land areas in the two hemispheres. In fact, the southernmost location in the Southern Hemisphere is at 56 deg South, while the northernmost location in the Northern Hemisphere is at about 72deg North. We note that the median return period of the Northern Hemisphere is even smaller (14 years) when only considering locations at latitudes below 56 deg North. Instead, we find that differences are consistent with the different distributions of the landmasses within the hemispheres relative to the storm tracks: a higher fraction of coastal areas experiences a relatively high cyclone frequency in the Northern compared to the Southern Hemisphere. We have added a sentence in the paper:

“Overall, concurrence probabilities in the Northern Hemisphere (median return period of 15 years) tend to be higher than in the Southern Hemisphere (23 years), which is consistent with the land distribution relative to the storm tracks (Fig. 2a-b) (e.g., 46% of the coast experiences more than 4 cyclones per month per 5deg spherical cap in the Northern Hemisphere compared to 18% in the Southern Hemisphere).”

Regarding the low probability of concurrent precipitation and meteorological tide extremes in equatorial regions, we have a sentence where we state: “Overall, extensive tropical regions with low cyclonic activity exhibit low concurrence probabilities (Fig. 1 and Fig. 2a,b).” The above is due to the fact that, although precipitation extremes are intense in these areas, they do not tend to concur with high meteorological tides. This is in line with what is explained in the paper when discussing Fig. 1, i.e. concurrent extremes are driven by cyclones, which are rare in the equatorial region. Of course, this does not imply that these regions are not exposed to flood risk, given that, as the referee mentions, extreme rainfall is important. However, the risk of flooding from concurring precipitation and meteorological extremes is limited due to the fact that high sea level and precipitation extremes tend to occur in different seasons in these regions, as shown by Fig. S3a. We have added a sentence (shown in bold) in the manuscript where we make this clear:

“We find in general that in areas with a low concurrence probability, coastal and inland meteorological extremes tend to happen in different seasons (see similar spatial distribution of blue areas in Fig. 1 and Supplementary Fig. S3a). This occurs, for example, in large areas of the tropics, indicating that the potential for flooding from concurrent meteorological extremes is limited. It may nevertheless be the case that there is a high risk of flooding from either hazard acting alone; our analysis here is of the compound flood hazard.”

6. It is not clear enough how the length of the concurrent extreme season was calculated. Does that figure mean that somewhere the concurrent extremes could last for 9 months?

Here, there is a misunderstanding. As we write in the slightly revised caption of Fig. 2, “The length of the concurrence season was defined as the shortest possible period within which 90% (range defined by the 5-95th percentiles) of concurrent extremes were observed.” Therefore, values of 9 months indicate that most (90%) of concurrent extremes were observed within a group of months that span 9 months. We have added an example in the text (in bold below) to help the reader:

“Fig. 2d shows the length of the compound season, i.e. the season within which 90% of coincident extremes occurs. For example, in Portugal concurrent extremes tend to occur mostly in December (Fig. 2c) and the season is about 3 months long (Fig. 2d), indicating that most of the concurrent extremes are observed around November-January. The longest season with concurrent extreme events is found along the eastern US coast (Fig. 2d), where they are caused by both TCs and ETCs [8] which hit the coast in different seasons.”

7. Another major issue is about the extreme thresholds defined. The current threshold is 0.95 or 0.997. Is that too “extreme”? In that case, there are not too many cases left when you selected the extreme events, particular for the compound extreme. Why not define 75% or 80% quantiles, probably not that “extreme”, but may cover more events which may significantly reduce the uncertainty caused by the scarce available data points.

First, we would like to mention that the methodology employed within the study has been widely tested and successfully used in previous works, e.g.:

- Bevacqua E, Maraun D, Voudoukas MI, Voukouvalas E, Vrac M, Mentaschi L, Widmann M. Higher probability of compound flooding from precipitation and storm

surge in Europe under anthropogenic climate change. *Science advances*. 2019 Sep 1;5(9):eaaw5531.

- Bevacqua, E., Vousdoukas, M. I., Shepherd, T. G., and Vrac, M.: Brief communication: The role of using precipitation or river discharge data when assessing global coastal compound flooding, *Nat. Hazards Earth Syst. Sci.*, 20, 1765–1782, <https://doi.org/10.5194/nhess-20-1765-2020>, 2020.

In the following, we explain in detail why the employed thresholds are not too “extreme”. First of all, it might be worth recalling that to allow for a robust estimation of the return periods, we fit a parametric bivariate probability density function only to pairs of high values. Applying a parametric model over the full range of values would run the risk of biasing the representation of the extreme tail by the bulk of the bivariate distribution where most data occur. We applied the parametric model to pairs of meteorological tides and precipitation that simultaneously exceed the individual 95th percentiles.

In this context, the 95th percentile was chosen carefully as a tradeoff between not selecting too few pairs and having a relatively high threshold. Selecting too few pairs would lead to very uncertain results (the concern of the referee), resulting in a spatially noisy map of the return periods. Employing too low thresholds would run the risk of biasing the representation of the extreme tail by data within the bulk of the bivariate distribution.

We nevertheless agree with the reviewer that, based on the 95th percentile thresholds, in a few locations (those with a low dependency between precipitation and meteorological tides), there might be not enough pairs. Therefore, in line with the referee’s comment and as we explain in the Methods, in these locations we lower the thresholds below the 95th percentile in order to ensure that a reasonable number N of pairs is selected and used for the fit of the bivariate distribution. Here, we fix $N=20$, however note that further tests showed that employing, e.g., $N=30$ or $N=40$ leads to virtually no differences in the return period estimation, highlighting the robustness of the method. (If interested, please see the almost identical return periods obtained based on $N=40$ and $N=20$ in a different work based on a dataset similar to the one used in the present study. Panels a in the first column of the Response Fig. 8, available at:

<https://www.nat-hazards-earth-syst-sci-discuss.net/nhess-2019-415/nhess-2019-415-AC1-supplement.pdf>). As an extra qualitative observation, note that the employed threshold leads to a map of the compound hazard which appears spatially smooth in space, indicating that uncertainty arising from limited sampling is small. Overall, the 95th percentile threshold in combination with the method above leads to selecting enough pairs.

Regarding the second threshold mentioned by the referee, i.e. 0.997, this is the percentile used to define extreme events. This threshold corresponds to precipitation (and meteorological tides) occurring on average once a year in the present climate. Within the context of extreme weather events, a 1-year return level is not too extreme (e.g., studies analysing present-day compound flooding employed similar or even higher thresholds; e.g., Couasnon et al., 2020; Bevacqua et al., 2020). For example, defining individual extremes as the 75 percentile would correspond to selecting precipitation events occurring on average every 4 days, which are not extreme and would not lead to characterising flooding due to concurring extremes (this threshold would result in very low return periods, that are not of interest for practitioners; while the chosen 0.997 threshold leads to a global median return period of 17 years). We agree that employing a very high threshold would lead to particularly large uncertainties; in fact, it would result in a return period map that is noisier in space

(despite being characterized by a similar large scale pattern of the compound hazard as shown in Fig. 1 (Bevacqua et al., 2020)).

For the reasons outlined above, we are convinced that 0.95 and 0.997 are reasonable thresholds.

Couasnon A, Eilander D, Muis S, Veldkamp TI, Haigh ID, Wahl T, Winsemius H, Ward PJ. Measuring compound flood potential from river discharge and storm surge extremes at the global scale and its implications for flood hazard. *Natural Hazards and Earth System Sciences*, 2019.

8. One important result is "latitudes above 40 degree north, compound meteorological extremes would become more than 2.5 times as frequent". I think such large difference sometime means little important information. For instance, based on the current thresholds used, the present concurrence probability may be around 0.0001, and for the future period, that probability is around 0.00025, which also means 2.5 times increased. But such event in fact is also impossible to occur, because the probability is to low. Thus, I would suggest the author consider lower thresholds to define individual extremes or the authors have some explanations on this issue.

Regarding the possibility of lowering the threshold, please see the comment above. Using a lower threshold would result in considering values of precipitation and storm surge that are not extreme and would therefore affect the validity of the results.

We accept the general point that, one should not only consider the changes in the return period, but also the reference value, as the referee implies with this comment. To guide the reader, we now write at the end of the presentation of the future changes:

"Overall, note that a large relative decrease in the return period does not necessarily imply a low return period in the future if the present-day return period is relatively high."

Having said that, with respect to the specific example identified by the referee, we would like to point out that the effects of an increase of "more than 2.5 times" (i.e., 2.6 times as stated in the section "Projections of concurrent extremes in precipitation and meteorological tide") in the concurrence of extremes would have tangible and important effects at latitudes above 40 degrees north.

First, to avoid misunderstanding, we note that if the referee refers to 0.0001 as the probability of an event occurring in a given day, this probability would correspond to a return period of $(1/0.0001)/365=27$ years. From the context of the comment, we understand that the referee refers to 0.0001 as the probability of an event occurring in a given year. Indeed, this probability corresponds to a return period of $(1/0.0001)=10000$ years, hence to an event that is very unlikely ("impossible") to occur, as the referee says.

However, in the case of the return periods at latitudes above 40 degrees north, the present-day probability of concurrent extremes is typically much higher than 0.0001. This is clear from Fig. 1, given that the return periods are in a range of 4-256 years ($\ll 10000$ years) in most regions above 40 degrees north (there are only a few exceptions at very high latitudes, where, however, the population density is very low; note that locations exposed to sea ice

above above the Polar circle are not considered in the new version of the manuscript in response to a comment of Referee 2). These return periods are in the range of typical design values and an increase of 2.6 times in the compound extreme frequency would lead to very important effects for practitioners. As an example, we consider two representative cases, i.e. the median return periods in two IPCC regions (data from the Supplementary Table 1):

- Even for the region with the highest return period or lowest probability of concurring extremes, i.e. Alaska/NW Canada, the present-day return period is 29 years (corresponding to a probability of $1/29=0.03$, which is much larger than 0.0001). Therefore, an increase of 2.6 times in the compound extreme frequency would lead to tangible differences in the occurrence of concurring extremes, given that it would correspond to a return period in the future of about 11 years. This implies that while in the present climate one expects concurrent extremes to occur, on average, every 29 years, in the future they would occur every 11 years. (The computation is based on the fact that an increase in the frequency of concurring extremes by 2.6 corresponds to a decrease of -61% in the return period, based on the equation provided at the end of the section "Return periods changes and robust changes").
- When considering a region with a lower return period or higher probability of concurring extremes, such as W N America, the present-day return period is 8 years, hence an increase by 2.6 of the compound extreme frequency would lead to certainly tangible change in the return period, which would be of about 3 years in the future.

9. The relative contribution of driver for each sub-area is interesting. But there is lack of in-depth analysis or discussion why there are large difference between sub-aeras about the drivers.

This comment makes us think that we should organise the section "Drivers of changes in concurrent meteorological extremes in coastal areas" in a better way to make clear that we first provide information on the relative contributions for each sub-area (the first two paragraphs), and then move on to the description of the physical drivers of such contributions and how they are linked to each other. In our view, referring to the individual 24 IPCC regions when describing the multiple physical processes behind the compound flooding meteorological drivers would make it difficult to follow the text. Therefore, while the information on the relative contribution in each sub-area (paragraph 2) is provided based on the IPCC regions (of interest for IPCC and practitioners), we provide the physical explanation of the drivers of such changes based on the spatial patterns of the changes, which are not necessarily aligned with the IPCC regions. We fully understand that we should better guide the reader within this context. Therefore, at the end of the second paragraph, we have added a sentence:

"We move on to examine these regional changes indirectly via discussing the physical processes shaping the large-scale changes in precipitation and meteorological tides, and how they are linked to each other. Changes in the dependences will be discussed in the next section relative to the uncertainty in the projections."

Regarding the discussion of the regional changes in the dependence, as we write in the first paragraph: "Changes in the dependence structure do not exhibit a clear large-scale spatial pattern and compensate each other at the global scale ($\Delta T_{dep.} = -1\%$), with approximately balanced fractions of coast with either increasing or decreasing dependency."

As we discuss in the section “Uncertainty in projections”, “The large variability in our projections of the dependence dynamics indicate that there is large uncertainty in how climate change could alter the concurrence of meteorological extremes in addition to the effects of changes in the marginal drivers.”.

Therefore, we have refrained from interpreting the dependence-driven changes from a physical point of view, given that any potential climate change signal in the dependence appears to be obscured by natural variability. Employing large ensemble model simulations would help in identifying a potential anthropogenic effect in the change of the dependence. We have added a sentence on this topic within the section “Uncertainty in projections” (the new part is shown in bold):

“The large variability in our projections of the dependence dynamics indicate that there is large uncertainty in how climate change could alter the concurrence of meteorological extremes in addition to the effects of changes in the drivers themselves; **large ensemble model simulations would be helpful to disentangle any anthropogenic-driven change in the dependence from natural variability, especially at regional and decadal scales (Deser et al., 2020).**”

Reviewer #2 (Remarks to the Author)

Bevacqua and co-authors have analysed projections of change in co-occurring extremes of coastal precipitation and meteorological tides at the global scale. Such compound events are relevant, because if present, would increase likelihood of extreme floods, with obvious consequences for coastal communities. Based on reanalysis data they show that such concurrent extremes are widespread, with a global average return period of 18 years. The two events are thus frequently linked, this return period is 20 times higher than would be expected for truly independent extremes. Using six CMIP5 models and a high-end scenario (RCP8.5) they then go on to show robust increases for many coasts, especially polewards of 40 N/S. These changes are dominated by changes in precipitation extremes, and in some regions a smaller change in meteorological tides or dependence. The topic of research is of interest, but I wonder if issues with TCs, or the lack of river/land processes have been sufficiently dealt with to warrant publication in this high-impact journal. In its current form, the manuscript paints too big a picture in the introduction. The results do not fill the knowledge gap painted. Hence, I recommend major revisions, either by modifying the analysis or writing a more fitting introductory text, before publication of this manuscript.

We thank the reviewer for the overall positive feedback on the manuscript, and for the constructive criticism that has led to a substantial improvement of the manuscript.

1- In the introduction the importance of compound flooding is noted, and add three TC-based examples (Harvey, Irma, Maria) are given. The fact that specifically TCs are unsatisfactorily simulated in CMIP models is a problem (which you note in the method section only). Though I realise the difficulty, I think this is something that needs to be solved before publication of the manuscript. One solution is to use HighResMIP data instead of CMIP data, at 25-50 km scale coupled models start to simulate hurricane force winds and more realistic TC spatial patterns. If this is unfeasible given computation costs, note early on that TCs are not included reliably in the investigation.

The datasets used in the study are the result of an unprecedented modelling effort for modelling storm surge and waves based on the output of global climate and ocean models. Therefore, as anticipated by the reviewer, the solution of using higher resolution data (e.g., HighResMIP) is unfeasible given computation costs in the storm surge and waves modelling. (Note that HighResMIP data is available only for a limited number of models and up to 2050, while we analyse climate change at the end of the century.)

We would like to also observe that, despite the existing limitation and the fact that the representation of the water level extremes could be improved by a higher resolution input data, 'smoothed tropical cyclones' are still present within CMIP5 models. Hence the passage of a tropical cyclone will still be associated with a peak of the water level, despite the peak will be underestimated. The issue of the underestimation of the tropical-cyclone-driven water level peaks is likely alleviated by the design of our analysis. In fact, we are not considering absolute storm intensities, but relative values. That is, we define extremes based on percentiles of the grid point distributions of meteorological tides and precipitation, which at least reduces the effect of biases in the magnitudes of the absolute values.

We discuss the above within the Methods:

“As discussed above, higher-resolution input data, such as from new ERA5 reanalysis and HighResMIP, could be used in future work to improve the representation of meteorological tides[1, 25]. Especially for tropical cyclones, the passage of a tropical cyclone will be associated with a meteorological tide peak that can be underestimated[37, 38, 25, 39]. However, such an underestimation is likely alleviated by the design of our analysis. In fact, we are not considering absolute storm intensities, but relative values. That is, we define extremes based on percentiles of the grid point distributions of meteorological tides and precipitation, which at least reduces the effect of biases in the magnitudes of the absolute values.”

However, we agree with the referee, as also stayed in the original version of the manuscript, that the results should be treated with caution in regions affected by tropical cyclones. In the introduction, we have provided examples caused by extratropical cyclones. Also, as suggested by the reviewer, we have moved the discussion on this topic from the methods section to the main text. Before presenting the results, we write:

“Given that higher-resolution input data could improve the representation of extreme events[1, 25, 36], especially for tropical cyclones (TCs)[37, 38, 25, 39], we improved the representation of TC-driven meteorological tides in the reanalysis based dataset. Storm surges caused by TCs were forced by dynamically downscaled atmospheric conditions and waves were corrected for TC effects based on satellite altimetry data (see Vousdoukas et al. [1] for more details). However, this procedure was not feasible for CMIP5-based simulations, therefore, despite the overall satisfactory representation of the compound hazard based on CMIP5 models in the present climate (Supplementary Fig. S1), the projected changes in regions subject to high TC activity should be interpreted with caution.”

2- In your abstract and introduction you talk about compound flooding and its importance. The paper then discusses the joint probability of extremes in meteorological tide and precipitation. These are not the same thing, as you note for the first time in the discussion (line 159). Please add these important details at the start of the paper too, overselling of what you are studying does not help the manuscript. A reference to Eilander et al. (2020, doi 10.1088/1748-9326/ab8ca6) is warranted as well, they provide a global analysis of compound flooding and do take into account riverine processes.

Please note that we were not trying to oversell, rather we discussed the interpretation of our results at the end in order to avoid repetition within the text. However, based on the comment of the referee, we now see and agree that it is certainly beneficial to discuss this topic earlier. We follow the suggestion and before presenting the results, we write:

“Following a methodology established in previous studies[8, 24, 10], we analyse the probability of concurring meteorological tide and precipitation extremes near the coast. Although our estimates should not be interpreted as an actual calculation of the flooding[24, 10, 26, 31], Bevacqua et al.[27] have shown that precipitation can provide a reliable estimate of compound flood potential from pluvial effects and in short- and medium-sized rivers, i.e. catchment size up to $5-10 \times 10^3 \text{ km}^2$, which is where the compound flood risk is the highest.”

“As discussed earlier, by using aggregated precipitation we do not aim at representing the compound flood potential in estuaries of long rivers (catchment $\geq 5-10 \times 10^3 \text{ km}^2$)[27], for which high discharges close to the coast are influenced by several processes over the catchment inland[27, 24, 41]. However, employing aggregated precipitation allows for considering local-rainfall-driven compound flood and, with some regional exceptions that will be discussed later, compound flood in small- and medium-size rivers, including small rivers not resolved by large-scale datasets[27].”

Of course, we still have a dedicated paragraph in the discussion, where we discuss the topic further, e.g., we write as in the original manuscript that the “estimates should not be interpreted as actual compound flood hazard”. Moreover, we discuss the new paper from Eilander et al., who focus on compound flooding in river estuaries in the present climate, both in the introduction and in the presentation of the results. In the presentation of the results, we compare our findings with those of Eilander et al.. In the introduction, we write:

“For example, Eilander et al.[28], using hydrodynamical modelling, focused on present-day compound flooding in river deltas highlighting that storm surge exacerbates 1-in-10 year flood levels in 64.0% of the analysed deltas worldwide.”

3- Why use an incomplete ERA-Interim dataset (1980-2014, not 1979-2019)? This should be updated. Better would be to use ERA5, the improved state-of-the-art reanalysis product from ECMWF.

Currently, reanalysis-based data of storm surges and waves are available only up to 2014, rendering not possible an extension to 2019. However, we believe that the results would not be particularly affected by an extension or shift of the dataset by four years. In any case, we make clear that the results are valid for the period 1980-2014, which should avoid any misunderstanding. We agree with the referee that using meteorological tides and precipitation from ERA5 would presumably lead to a better representation of extreme events, but runs for storm surge and waves are currently unavailable for ERA5. However we can see no reason to believe that our ERA-Interim based results are unreliable, given that ERA-Interim has been so widely used. As described for tropical cyclones, potential differences in the water levels arising from resolution are likely alleviated by the design of our analysis defining extremes based on percentiles of the grid point distributions of meteorological tides and precipitation, which at least reduces the effect of biases in the magnitudes of the absolute values. We have now referred explicitly to the possibility of using ERA5 in the limitation section:

“As discussed above, higher-resolution input data, such as from new ERA5 reanalysis and HighResMIP, could be used in future work to improve the representation of meteorological tides[1, 25].”

- Lines 39-41: This sentence is not clear, especially "forced with reanalysis data for the present and with CMIP5 GCM climate projections up to...". Maybe rewrite along the lines "forced with reanalysis data for the observed past, and with CMIP5 GCM projections for estimates of future climates."

Thank you. We have changed the text.

- *Line 60: I'd note cyclones normally travel away from the coast near Central America.*

Thanks, we adapted the text: "Off the west coast of Central America and Mexico, TCs are also frequent but they usually travel away from the coast near Central America (Fig. 2b), which results in somewhat higher joint return periods (8-16 years)."

- *Line 71: Why change the name to 'inland extremes'?*

We have changed to "meteorological tide and precipitation extremes". Note that earlier on in the manuscript, we also use "coastal and inland meteorological extremes". We are unsure on whether the reviewer is suggesting to avoid the term inland at all when referring to precipitation extremes. We would be happy to change the text further if that could help the readability of the manuscript.

- *Line 182: Please add why emergency response is different for floods by only meteorological tide, or floods due to both. I doubt response is hampered because of rain.*

We refer here to the fact that concurrent but not hydrologically interacting storm surges and rainfall extremes, i.e. not leading to "compound flooding" but to concurrent impacts in isolation, may, e.g., limit the ability to respond to emergency, and amplify the impacts that the two hazards would have caused if they occurred in isolation. Civil protection may be designed to cope with one hazard at a time. If the concurrence probability increases, the capacity of civil protection may not be sufficient to cope with both disasters at the same time. We have rephrased:

"Neglecting compound flood hazard and changes therein might leave several parts of the global coastline insufficiently protected. Concurrent extreme events may be more challenging compared to individual coastal or inland flooding events also for taking emergency response action, e.g. they may amplify impacts through overloading rescue teams[58, 59]. The presently reported increasing frequency of such concurrent hazards highlights the need for better planning of emergency response and designing efficient protective structures."

- *Figure1: Why are large parts of northern Russian coast/Canada/Greenland not modelled?*

As discussed in the Methods of the original version of the manuscript, our ocean models do not resolve properly ice-water interaction, therefore results should be interpreted with caution at areas with substantial ice presence. For that reason, we decided not to include those ice-dominated parts of the global coastline in our analysis. This approach has also been followed by other studies (e.g., Hemer et al., 2013; Vousdoukas et al., 2018). In our case, we have now removed all the coastal locations above the Polar circle typically exposed to sea-ice in winter, resulting in a few minor changes in some of the statistics presented within the manuscript, e.g., the global median return period is 17 years in the new version of the manuscript rather than 18 years. For clarity, in the Methods we have written:

"We have not considered the locations above the Polar circle that are affected by ice related processes not properly resolved by our ocean models (Vousdoukas et al., 2018)"

Hemer MA, Fan Y, Mori N, Semedo A, Wang XL. Projected changes in wave climate from a multi-model ensemble. *Nature climate change*. 2013 May;3(5):471-6.

Vousdoukas MI, Mentaschi L, Voukouvalas E, Verlaan M, Jevrejeva S, Jackson LP, Feyen L. Global probabilistic projections of extreme sea levels show intensification of coastal flood hazard. *Nature communications*. 2018 Jun 18;9(1):1-2.

- Figures: Given that we are talking about wet extremes (rain and tide), I suggest swapping you color scales to show high co-occurrence in blue.

We chose to use red colours for locations with higher or increasing hazard probability, in line with the overall interpretation of red as indicative of high risk. However, we would be happy to change the colour if the editor shares the same opinion as the reviewer. Please note that changing the colour of Fig.1 would require, for consistency, also changing the colours in Figs. 2a, 2b, 3, 4, S3a, S4, S5.

- Figure 2c: Does colouring by season rather than calendar month provide more information? Large differences between northern and southern hemisphere are not a dynamical feature, but result of course from differences in solar insolation.

We understand that the reviewer is suggesting to use four colours, one per season. We have considered this possibility, but we refrained from doing that, given that the four seasons are not a standard everywhere in the world, especially in the tropics. Employing 12 months also allows for a more continuous colour palette.

- Figure 5: Can you scale the circles by the absolute uncertainty in projected changes, such that large circles show larger uncertainties? If absolute uncertainty is low, the source of that uncertainty is less relevant.

We find this a minor point since anyway we discuss the robustness of the changes when describing Fig. 3 and 4 in the previous sections. In the above figures, large model disagreement and robust changes are highlighted via magenta and grey points. So, unless the Editor has a different view, we would prefer not to change figure 5. The main reason is that within this section and Fig. 5, we mainly focus on the relative contributions to the uncertainties, and most importantly, we highlight the novel result that the dependence between the drivers leads to a large fraction of the uncertainty in any particular region. Given this specific target, we have changed the title of the subsection from "Uncertainty in projections" to "Relative contributions of the drivers to the uncertainty in the projections".

26th Aug 20

Dear Dr Bevacqua,

Your manuscript titled "Global projections of changes in meteorological drivers of compound coastal flooding" has now been seen by our reviewers, whose comments appear below. In light of their advice I am delighted to say that we are happy, in principle, to publish a suitably revised version in Communications Earth & Environment under the open access CC BY license (Creative Commons Attribution v4.0 International License).

We therefore invite you to revise your paper one last time to address the remaining concerns of our reviewers. At the same time we ask that you edit your manuscript to comply with our format requirements and to maximise the accessibility and therefore the impact of your work.

EDITORIAL REQUESTS:

Please review our specific editorial comments and requests regarding your manuscript in the attached "CommsEarth Final revisions information checklist". Please outline your response to each request in the right hand column.

SUBMISSION INFORMATION:

In order to accept your paper, we require the files outlined in the attached "CommsEarth Final submission file checklist.pdf"

OPEN ACCESS:

Communications Earth & Environment is a fully open access journal. Articles are made freely accessible on publication under a [CC BY license](http://creativecommons.org/licenses/by/4.0) (Creative Commons Attribution 4.0 International License). This license allows maximum dissemination and re-use of open access materials and is preferred by many research funding bodies.

For further information about article processing charges, open access funding, and advice and support from Nature Research, please visit <https://www.nature.com/commsenv/about/open-access>

At acceptance, the corresponding author will be required to complete an Open Access Licence to Publish on behalf of all authors, declare that all required third party permissions have been obtained and provide billing information in order to pay the article-processing charge (APC) via credit card or invoice.

Please note that your paper cannot be sent for typesetting to our production team until we have received these pieces of information; therefore, please ensure that you have this information ready when submitting the final version of your manuscript.

[link redacted]

Best regards,

Heike Langenberg, PhD

Chief Editor
Communications Earth and Environment

On Twitter: @CommsEarth

REVIEWERS' COMMENTS:

Reviewer #1 (Remarks to the Author):

my concerns have been well addressed.

Reviewer #2 (Remarks to the Author):

I thank the authors for their careful and detailed response to my remarks and questions. I am happy with the changes they have made. My advise is thus that this manuscript is accepted for publication. Congratulations to the authors on this interesting paper.